# An Exploration of Rural–Urban Residence on Self-Reported Health Status with UK Cancer Survivors Following Treatment: A Brief Report

**David Nelson [1,2,\*], Ian McGonagle [3], Christine Jackson [3] and Ros Kane [3]**

1   Lincoln International Institute for Rural Health, College of Social Science, University of Lincoln, Brayford Pool, Lincoln LN6 7TS, UK
2   Macmillan Cancer Support, London SE1 7UQ, UK
3   School of Health and Social Care, University of Lincoln, College of Social Science, Brayford Pool, Lincoln LN6 7TS, UK
\*   Correspondence: dnelson@lincoln.ac.uk; Tel.: +44-(0)1522-837343

**Abstract:** Objective: To explore the effect of rural–urban residence on the self-reported health status of UK cancer survivors following primary treatment. Design: A post-positivist approach utilizing a cross-sectional survey that collected data on demographics, postcode and self-reported health status. Methods: An independent samples *t* test was used to detect differences in health status between rural and urban respondents. Pearson's $\chi^2$ was used to control for confounding variables and a multivariate analysis was conducted using Stepwise linear regression. Setting: East Midlands of England. Participants: Adult cancer survivors who had undergone primary treatment in the last five years. Participants were excluded if they had recurrence or metastatic spread, started active oncology treatment in the last twelve months, and/or were in receipt of palliative or end-of-life care. Main outcome: Residence was measured using the UK Office for National Statistics (ONS) RUC2011 Rural–Urban Classifications and Health Status via the UK ONS self-reported health status measure. Ethics: The study was reviewed and approved (Ref: 17/WS/0054) by an NHS Research Ethics Committee and the Health Research Authority (HRA) prior to recruitment and data collection taking place. Results: 227 respondents returned a questionnaire (response rate 27%). Forty-five percent (*n* = 103) were resident in a rural area and fifty-three percent (*n* = 120) in an urban area. Rural (4.11 ± 0.85) respondents had significantly (*p* < 0.001) higher self-reported health statuses compared to urban (3.65 ± 0.93) respondents (MD 0.47; 95% CI 0.23, 0.70). Conclusion: It is hoped that the results will stimulate further work in this area and that researchers will be encouraged to collect data on rural–urban residency where appropriate.

**Keywords:** rural health; urban health; health status; cancer survivors; United Kingdom

## 1. Introduction

Cancer is a leading cause of death globally and accounted for nearly ten million deaths in 2020 [1]. The burden of cancer continues to grow, exerting considerable physical, emotional and financial pressure on people living with and affected by cancer, as well as on health and social care systems around the world [2]. The Global Burden of Disease (GBD) study has identified large heterogeneities with regard to cancer care and survival which can be attributed to exposures to risk factors, lifestyles, access to treatment and screening, as well as different economic and geographic settings [3].

The majority of the cancer survivorship literature with a focus on geography and specifically with regard to rurality tends to be from Australia and North America [4–20]. Whilst these geographies are significantly larger than the UK and what constitutes 'rural' in the UK might be very different than in North America or Australia, it has been well documented that cancer survivors in rural settings experience a number of additional

challenges compared to their urban counterparts [13,16,19–23]. Examples include having to travel long distances for treatment, less access to bespoke support and emotional and physical isolation. Additionally, research has highlighted the benefits of rural living and 'green spaces' in improving physical and mental health and there are a number of benefits of rural living for cancer survivors [24,25]. For example, rural communities frequently value close relationships with family and friends, community members and religious institutions [26,27], which constitute significant sources of social support [24] vital to coping with or minimizing emotional distress when experiencing a traumatic life event, such as a cancer diagnosis. Butow et al. [21] maintain that this could be a potential reason for differences and that rural populations might be less inclined to ask for help; interestingly, recent research in Australia highlighted that people affected by cancer in rural areas were less likely to report higher levels of distress compared to those from urban areas [6].

Our recent research has looked specifically at the role of rural–urban residency on the self-management and cancer-related self-efficacy of UK cancer survivors [28,29]. In the UK, almost a fifth (10.82 million) of the total population (56.39 million) reside in rural areas [30]. It is therefore important to understand the experiences and health outcomes of cancer survivors who reside in both rural and urban areas. However, we still do not understand the impact of geography on health status as it directly relates to UK populations who have completed primary treatment for cancer. This research aimed to address that gap by exploring the impact of rural–urban residence on self-reported health status of cancer survivors who were post-treatment in the East Midlands region of England.

## 2. Methods

### 2.1. Study Design

The study utilized a post-positivist approach via a cross-sectional self-completion postal survey that was administered to adult cancer survivors who had completed primary treatment.

The Strengthening the Reporting of Observational studies in Epidemiology (STROBE) Checklist for cross-sectional studies was adhered to when reporting this study [31].

The research was approved by a School of Health and Social Care Ethics Committee (Ref: 12/02/17) as well as by a National Health Service (NHS) Research Ethics Committee (REC) and the Health Research Authority (HRA) (Ref: 17/WS/0054; IRAS Project ID: 204679). NHS approvals had to be in place, given the use of NHS systems and staff to support the recruitment of eligible participants. Furthermore, confirmation of capacity and capability to deliver the study was authorized by both participating NHS trusts' Research and Development (R&D) departments.

### 2.2. Setting

The study setting was the East Midlands of England which has been considered a microcosm of the UK in terms of demographics, urban-rural dynamics and deprivation [32].

### 2.3. Questionnaire Design

The questionnaire was developed by the research team using the extant literature and overseen by a project Steering Group that had representation from clinical oncology professionals, senior staff from a UK cancer support charity and health professionals from the local NHS Clinical Commissioning Group (CCG), as well as an individual with lived cancer experience. The questionnaire was piloted with a patient and public involvement (PPI) group prior to finalising the content. It collected data on demographics, self-reported health status and postcode to ascertain rural–urban residence. Clinical data on cancer type was provided anonymously by the participating NHS trusts.

#### 2.3.1. Demographics

The survey collected data on a range of demographics: age, gender, living arrangement, marital status, employment status, qualifications and income.

2.3.2. Health Status

For health status, we used the UK Office for National Statistics (ONS) measure in which participants were asked the question 'How is your health in general?' and they could rate their health as 'Very Good', 'Good', 'Fair', 'Poor' or 'Very Poor'. This method has been used in the UK census [33] and was in line with existing research exploring rural–urban differences in health behaviours and health status with American cancer survivors that were also asked to self-report their health status [8,9].

2.3.3. Rural–Urban Residence

Respondents were asked for their postcode and rural–urban residence was defined based on the UK ONS RUC2011 Rural Urban Classifications [34] using the ONS postcode directory lookup tool (https://onsdigital.github.io/postcode-lookup/ last accessed on 1 May 2022) which has been recommended for statistical analyses by the UK Department for Environment, Food and Rural Affairs (Defra) [35]. Participants can be assigned to one of four urban categories or six rural categories. To allow for comparison between the two groups, respondents were assigned to a dichotomous variable that categorized them as rural or urban. This approach of using official statistics to define rural–urban residence has been adopted internationally in other cancer research studies in high income settings [8,9,13]. The ONS measure for Index of Multiple Deprivation (IMD) was also assigned utilizing postcode data [36].

*2.4. Participant Eligibility*

Participants were included if they were aged 18 years and over, had a confirmed cancer diagnosis and had undergone treatment in the last five years, and excluded if they had evidence of recurrence or metastatic spread, started active oncology treatment in the last twelve months and/or were currently being treated for palliative or end-of-life care.

*2.5. Sample Size Calculation*

The first author (DN) worked with an experienced statistician to calculate the sample size and a letter of support was provided to the ethics committee outlining how the sample size was calculated. The calculation was performed for an independent samples (rural and urban) *t* test in relation to the outcome measure. The final calculation allowed for a 20 percent difference between scores, assumed a statistical significance level of 0.05 and a test with 95 percent power giving a required sample of 417. In line with similar cancer survivorship research in the West Midlands of England that also used a self-completion postal questionnaire [37,38], the sample size was doubled as it was anticipated that 50 percent of participants would respond. Therefore, 834 participants that met the above eligibility criteria were identified and sent a questionnaire.

*2.6. Recruitment*

Access to the sample population was sought via cancer centre staff at two acute NHS Trusts who acted as gatekeepers to the study population. These were both based in the East Midlands of England, one which covers a sparse and rural county and another with a high proportion of urban dwellers. The managers of both cancer centers as well as their lead cancer nurse specialists (CNSs) were briefed on the eligibility criteria and confirmed that they could identify and recruit potential participants via their patient database on behalf of the research team. An information analyst at each trust led the identification of potential participants using their patient database. The research team did not have access to identifiable patient information. A random sample of 834 eligible participants (417 at each NHS site) were identified and sent a printed research pack in the post that included an NHS-branded invitation from the lead CNS at each site, a participant information sheet, questionnaire and a freepost return envelope to the lead researcher's work address. These materials were designed to tell the participant more about why they were invited, the purpose of the research and the conditions of taking part. The draft study materials were

piloted with five volunteers who had lived experience of cancer prior to seeking ethical approval. The research packs were sent out in June 2017 at one site and in September 2017 at the other participating NHS site.

*2.7. Consent*

It was made clear on the information sheet and questionnaire that by completing and returning the questionnaire, the participant was giving their consent to take part and the conditions were outlined in the information sheet. Participants could self-select to take part after reading the materials and it was made clear that participation was entirely voluntary. We did not have access to the characteristics of non-responders as recruitment was conducted via the participating NHS trusts on our behalf.

*2.8. Analysis*

Descriptive statistics were used to characterize the data and an independent samples *t* test was first used to assess for significance between rural and urban respondents. Pearson's $\chi^2$ was used to assess for confounding variables. Finally, a multivariate analysis was conducted using a stepwise linear regression whilst controlling for confounding variables. We used the forward selection method, which involves using several models testing the addition of each independent variable on the dependent variable (health status) and repeating the process until there is no improvement with statistical significance. With this method, non-significant predictors are excluded from the models. The results were considered significant if $p < 0.05$. The data were analysed in SPSS software (Ver. 25).

## 3. Results

A total of 227 respondents (response rate of 27%) returned a questionnaire, and the mean overall age was 66.86 years ±11.22 (range 26–90). Fifty-two percent (*n* = 119) were female and forty-eight percent (*n* = 108) were male. Forty-five percent (*n* = 103) were resident in a rural area and fifty-three percent (*n* = 120) in an urban area.

When comparing between urban and rural participants in our sample, there are some important differences to highlight. More urban participants reported living alone (25% vs. 9%) and being single when compared to rural partipants (32% vs. 13%). Furthermore, twenty-five percent (*n* = 30) of urban respondents reported having no qualification compared to eight percent (*n* = 8) of rural respondents.

In terms of health status, eighty percent (*n* = 82) of rural respondents reported their health as very good or good compared to sixty-one percent (*n* = 73) of urban respondents. Ten percent (*n* = 12) of urban participants self-reported their health as poor or very poor compared to 4 percent (*n* = 4) of rural participants. Finally, it should also be noted that less rural respondents self-reported their health as fair compared to urban respondents. Full rural and urban participant characteristics are reported on in Table 1.

**Table 1.** Rural–Urban Comparison of Participants.

| Characteristic | | Rural Total n = 103 | Urban Total n = 120 |
|---|---|---|---|
| | | n (%) | n (%) |
| Age | 25–44 | 3 (2.9) | 4 (3.3) |
| | 45–54 | 12 (11.7) | 15 (12.5) |
| | 55–64 | 24 (23.3) | 26 (21.7) |
| | 65–74 | 42 (40.8) | 48 (40.0) |
| | Over 75 | 22 (21.4) | 27 (22.5) |
| Gender | Female | 62 (60.2) | 57 (47.5) |
| | Male | 41 (39.8) | 63 (52.5) |
| Living arrangements | Partner/Spouse/Family/Friends/Nursing home | 92 (89.3) | 90 (75.0) |
| | Alone | 9 (8.7) | 30 (25.0) |

**Table 1.** *Cont.*

| Characteristic | | Rural Total n = 103 | Urban Total n = 120 |
|---|---|---|---|
| | | n (%) | n (%) |
| Marital status | Married or living with partner | 89 (86.4) | 82 (68.3) |
| | Single/Divorced/Separated/Widowed | 13 (12.7) | 38 (31.7) |
| Employment status | Employed | 21 (20.4) | 30 (25.0) |
| | Not Employed | 3 (2.9) | 9 (7.5) |
| | Retired | 69 (67.0) | 75 (62.5) |
| | Other | 9 (8.7) | 6 (5.0) |
| Qualifications ** | Professional | 30 (29.1) | 30 (25.0) |
| | Qualification | 19 (18.4) | 20 (16.7) |
| | Degree or Higher Degree | 24 (23.3) | 26 (21.7) |
| | A level or equivalent GCSE/O Levels or equivalent | 35 (34.0) | 43 (35.8) |
| | No qualifications | 8 (7.8) | 30 (25.0) |
| Annual household income | £0–14,999 | 23 (22.3) | 31 (25.8) |
| | £15–24,999 | 19 (18.4) | 36 (30.0) |
| | £25–49,999 | 36 (35.0) | 35 (29.2) |
| | Over £50,000 | 12 (11.7) | 8 (6.6) |
| Primary Cancer Type | Breast | 39 (37.9) | 34 (28.6) |
| | Urological | 22 (21.4) | 30 (25.2) |
| | Skin | 8 (7.8) | 10 (8.4) |
| | Head and Neck | 7 (6.8) | 6 (5.0) |
| | Gynecological | 6 (5.8) | 4 (3.4) |
| | Lower Gastrointestinal | 13 (12.6) | 16 (13.4) |
| | Hematological | 4 (3.9) | 6 (5.0) |
| | Upper Gastrointestinal | 3 (2.9) | 8 (6.7) |
| | Other | 1 (1.0) | 5 (4.2) |
| Health Status | Very Good or Good | 82 (79.6) | 73 (60.9) |
| | Fair | 17 (16.5) | 35 (29.1) |
| | Poor or Very Poor | 4 (3.9) | 12 (10.0) |

Note: Column percentages are reported. Percentages may not total 100% due to missing values. ** Percentages add to more than 100% because participants could select more than one option.

Firstly, the independent samples *t* test revealed that rural (4.11 ± 0.85) respondents had significantly ($p < 0.001$) higher self-reported health status compared to urban (3.65 ± 0.93) respondents (MD 0.47; 95% CI 0.23, 0.70). Pearson's $X^2$ test revealed that living arrangement (9.768, $p = 0.002$ *), marital status (11.155, $p = 0.001$ *) and qualifications (11.886, $p = 0.003$ *) were all significantly associated with rural–urban residence and so, were entered into our stepwise linear regression model with rural–urban residence and deprivation to adjust for their effect (Table 2).

Turning to the multivariate analysis in Table 2, the first model showed that rural–urban residence was a highly significant ($p = 0.000$) predictor of health status. In models two, three, and four, with the inclusion of additional significant predictors, rural–urban residence remained a significant predictor of self-reported health status. Qualifications were not a significant predictor and as such, were excluded from the models. Model 4, which included marital status, living arrangements and deprivation as covariates, was the best fit, although the adjusted r² (0.117) was only slightly larger than model 3 (0.103). Deprivation, living arrangement and marital status were also significant predictors in this model when controlling for confounding variables. Rural–urban residency still appeared to have a strong effect on health status, but it was marital status which was the most highly significant predictor in model 4. Whilst the adjusted r² for model 4 could be considered low there was a notable increase from model 1 and 2, suggesting that this model was the best fit, although further research that considers additional covariates is required.

**Table 2.** Multiple Predictors of Health Status Using Stepwise Linear Regression.

| | Health Status | | | | |
|---|---|---|---|---|---|
| **Model 1** | **B** | **SE B** | **β** | **t** | **p** |
| Constant | 3.701 (3.541, 3.860) | 0.081 | | 45.752 | 0.000 |
| Rural–Urban | 0.430 (0.195, 0.666) | 0.119 | 0.239 | 3.603 | 0.000 |
| Adjusted R$^2$ | 0.053 | | | | |
| **Model 2** | | | | | |
| Constant | 3.427 (3.179, 3.675) | 0.126 | | 27.254 | 0.000 |
| Rural–Urban | 0.356 (0.119, 0.594) | 0.121 | 0.198 | 2.958 | 0.009 |
| Marital Status | 0.400 (0.119, 0.681) | 0.142 | 0.188 | 2.810 | 0.013 |
| Adjusted R$^2$ | 0.082 | | | | |
| **Model 3** | | | | | |
| Constant | 3.125 (2.766, 3.474) | 0.177 | | 17.671 | 0.000 |
| Rural–Urban | 0.318 (0.081, 0.555) | 0.120 | 0.177 | 2.643 | 0.009 |
| Marital Status | 0.357 (0.077, 0.637) | 0.142 | 0.168 | 2.515 | 0.013 |
| Deprivation | 0.057 (0.010, 0.103) | 0.024 | 0.159 | 2.405 | 0.017 |
| Adjusted R$^2$ | 0.103 | | | | |
| **Model 4** | | | | | |
| Constant | 3.271 (2.900, 3.643) | 0.189 | | 17.349 | 0.000 |
| Rural–Urban | 0.329 (0.094, 0.564) | 0.119 | 0.183 | 2.755 | 0.006 |
| Marital Status | 0.806 (0.304, 1.308) | 0.255 | 0.379 | 3.166 | 0.002 |
| Deprivation | 0.055 (0.009, 0.101) | 0.023 | 0.155 | 2.369 | 0.019 |
| Living Arrangement | −0.593 (−1.145, −0.041) | 0.280 | −0.252 | −2.116 | 0.035 |
| Adjusted R$^2$ | 0.117 | | | | |

Notes: Figures in brackets refer to 95% confidence intervals. Outcome Health Status: Very Poor = 1, Poor = 2, Fair = 3, Good = 4, Very Good = 5; Residence: Urban = 0 and Rural = 1; Marital Status: Widowed/Single/Divorced/Separated = 0 Married/Civil Partnership = 1; Deprivation: 1 = Most Deprived through to 10 = Least Deprived. Living Arrangement: 0 = Live alone and 1 = Partner/Spouse/Family/Friends.

## 4. Discussion

The research highlighted that in this sample, rural respondents had significantly higher self-reported health statuses compared to their urban counterparts. To the best of our knowledge, this is the first study of its kind that has compared the self-reported health statuses of rural and urban post-treatment cancer survivors in the UK. The findings were at odds with American research where rural participants were more likely to self-report fair and poor health [8,9]. However, their sample was considerably larger and those from rural areas had lower levels of education and health insurance compared to their urban counterparts [9]. Model 4, which included rural–urban residence, marital status, living arrangements and deprivation, was the best predictor of self-reported health status. Marital status was also a highly significant predictor of health status, indicating the importance of social support on health and wellbeing following cancer treatment. That being said, these results from the multivariate analysis should be interpreted with a certain level of caution given the low adjusted r$^2$. However, it is hoped that the results will stimulate further work in this area and that researchers will be encouraged to collect data on rural–urban residency when appropriate. This can be facilitated by asking participants for their postcode and

cross-referencing with official statistics, as was the case in this study. This means that the amount of personal data that is required is minimal.

The literature maintains that cancer survivors in rural areas tend to be more stoic with regards to their health [39] and are less likely to report high levels of distress [6] compared to their urban counterparts, which could account for some of the differences in health status in this study. Research by McNulty and Nail [10] found that rural participants advocate for themselves, their diagnosis, their survivorship and improved health care, and also have higher levels of trust within their local communities [6] which could subsequently impact positively on health status. Our other research with UK cancer survivors also suggests that people from rural areas have greater confidence to self-manage when compared to their urban counterparts [28]. Further qualitative work is warranted with diverse samples from a range of rural and urban areas to understand why health status and recovery might differ between rural and urban populations who have completed treatment for cancer.

*Limitations*

A limitation of our study was that we did not have access to the details of non-responders and so could not make any conclusions about those who decided not to take part and whether they came from a rural or urban area. A further limitation was that the required sample size of 417 was not reached; however, the sample of 227 still offered a good split in terms of rural and urban respondents to facilitate a comparative analysis, although not knowing the details of non-responders meant that we could not adjust our analyses to account for non-response We chose to dichotomize rural–urban residency to fit with our approach to analysis, which meant that we might not have been able to distinguish between some of the potential differences between the four urban and six rural categories as designated by the UK ONS. Finally, the use of self-report measures also raises concerns around response bias, which needs to be considered when interpreting the findings.

## 5. Conclusions

This is the first analysis of self-reported health status between rural and urban cancer survivors in the UK who have completed primary treatment. Much of the existing evidence comes from North America and Australia and this brief report offers a welcome addition to the extant literature within a UK context in which there is a notable gap in the evidence. In this study rural respondents had significantly greater self-reported health status when compared to their urban counterparts which could suggest that there are positives to rural living that can support people with their recovery and longer-term survivorship. That said, we know that rural cancer survivors have unique psycho-social needs that often go unmet and face a number of additional barriers when it comes to receiving optimal care compared to their urban counterparts. Despite our somewhat positive findings in relation to rural respondents self-reporting better health statuses, it is critical that rural oncology providers do not rely on cancer survivors to co-ordinate their own care and that rural cancer nurses work collaboratively with survivors and other cancer support professionals at all stages of the pathway to facilitate a positive patient experience that will support their recovery.

**Author Contributions:** Conceptualization, D.N., I.M., C.J. and R.K.; methodology, D.N. and R.K.; software, D.N.; validation, D.N. and R.K.; formal analysis, D.N.; investigation, D.N., I.M., C.J. and R.K; data curation, D.N.; writing—original draft preparation, D.N. and R.K.; writing—review and editing, D.N., I.M., C.J. and R.K; supervision, I.M., C.J. and R.K.; project administration, D.N.; funding acquisition, R.K. All authors have read and agreed to the published version of the manuscript.

**Funding:** This research was funded by Macmillan Cancer Support and the University of Lincoln.

**Institutional Review Board Statement:** The research was reviewed and approved by a University of Lincoln Ethics Committee (Ref: 12/02/17) and a National Health Service (NHS) Research Ethics Committee (Ref: 17/WS/0054).

**Informed Consent Statement:** Informed consent was obtained from all participants involved in the study.

**Data Availability Statement:** The anonymized data that support the findings of this article are available from the corresponding author (D.N.) upon reasonable request. Restrictions apply to the availability of these data due to the conditions of the ethical approval.

**Acknowledgments:** We would like to acknowledge the participants who took part in the study and the members of the Project Steering Group who directed it.

**Conflicts of Interest:** The authors declare no conflict of interest.

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
