# Peer review of "An Exploration of Rural–Urban Residence on Self-Reported Health Status with UK Cancer Survivors Following Treatment: A Brief Report"

_nursrep, doi:10.3390/nursrep12030056_

Round 1
Reviewer 1 Report
Dear authors, congratulations on the improvement of the article. However, it still needs improvements for its publication.
Abstract: In methodology, it does not specify a research paradigm, the validation of instruments, or how ethical considerations were respected in the study.
Introduction: has improved.
Methodology: It is necessary to declare the research paradigm used to review the study's coherence and the article's writing.
It is also essential to establish how the applied questionnaire was developed.
Regarding data collection, there is a lack of information regarding whether there were refusals to participate and how the information was protected.
In the data analysis, it should be explicit how the participants' data were protected. For example, in the event of non-response, participation was insisted on or considered a refusal. In addition, you must specify how you ensured compliance with the ethical considerations in each methodology section.
Results: These must relate to the methodology. For example, there is talk of Models which are not previously named in the introduction or the method.
Discussion: It needs to be improved. Only two articles from the last five years are discussed; the rest are not original articles or are more than five years old.
Likewise, it is scarce compared to many results presented.
It is suggested to use subtitles.
Conclusions: They are not clear. What is the real contribution of this research to Nursing, and if it responds to the stated objective?
Author Response
Reviewer 1
Dear authors, congratulations on the improvement of the article. However, it still needs improvements for its publication.
Response: Thank you very much for your positive comments and for continuing to review our article, we really appreciate your time and constructive feedback that are helping to improve the quality of our submission to Nursing Reports. We do hope you will take into account that this article is classified as a ‘Brief Report’ and therefore we are limited as to the level of detail that we can go in to when compared to a full research article, but we have aimed to address your comments as best we can previously and during this new round of peer review.
Abstract: In methodology, it does not specify a research paradigm, the validation of instruments, or how ethical considerations were respected in the study.
Response: Thank you, under design in the abstract, we have re-inserted that a post-positivist approach was utilized in response to the research paradigm that underpinned the study. Regarding the validation of instruments, we used UK Office for National Statistics (ONS) measures to collect data on health status and rural-urban residence, and this is reported under the main outcomes in the abstract. These measures are widely used and are survey instruments that are part of the population level census conducted in the UK. We have a statement in the abstract saying that the study was reviewed and approved by several ethics/governance bodies such as an National Health Service (NHS) Research Ethics Committee (Ref: 17/WS/0054) and the Health Research Authority (HRA) prior to recruitment and data collection. Furthermore, in the body of the text we go into more detail in that Research and Development (R&D) departments from the participating hospital trusts had to be consulted and to confirm capacity and capability to support the research following the ethical approvals.
Introduction: has improved.
Response: Thank you based on your previous comments we have added some more up to date references to signpost to the salient literature in this area.
Methodology: It is necessary to declare the research paradigm used to review the study's coherence and the article's writing.
Response: We have now re-inserted the text that the study utilized a post-positivist approach.
It is also essential to establish how the applied questionnaire was developed.
Response: Under questionnaire design we have added some text stating that the questionnaire was developed by the research team using the existing academic literature and overseen by a project Steering Group that had representation from clinical oncology professionals, senior staff from a UK cancer support charity and health professionals from the local NHS Clinical Commissioning Group (CCG) as well as an individual with lived cancer experience.
Regarding data collection, there is a lack of information regarding whether there were refusals to participate and how the information was protected.
Response: Participants self-selected to take part and it was made clear that their participation was voluntary. We did not have access to the details of people who decided not to take part as this was done via the recruiting NHS trusts on our behalf. We have added a sentence in relation to this under 2.7 consent.
In the data analysis, it should be explicit how the participants' data were protected. For example, in the event of non-response, participation was insisted on or considered a refusal. In addition, you must specify how you ensured compliance with the ethical considerations in each methodology section.
Response: Access to the sample population was sought via Cancer Centre staff at the participating NHS trusts who acted as gatekeepers to the study population. An information analyst at each trust led on the identification of potential participants and not the research team. We have stated this under the recruitment section. Furthermore, participation was not insisted, participants could self-select to take part after reading the materials that were sent to them, so participation was voluntary, this is reported on under the consent heading. We did not have access to the data of non-responders as recruitment was done via NHS staff and this is reported on twice in the manuscript.
Re compliance - the research had to be approved by several bodies such as a School of Health and Social Care Ethics Committee at the host academic institution (Ref: 12/02/17) as well as by a National Health Service (NHS) Research Ethics Committee (REC) and the Health Research Authority (HRA) (Ref: 17/WS/0054; IRAS Project ID: 204679). NHS approvals had to be in place given the use of NHS systems and staff to support the recruitment of eligible participants. Furthermore, confirmation of capacity and capability to deliver the study was authorized by both participating NHS trusts Research and Development (R&D) departments. This is all reported on under 2.1 study design.
Results: These must relate to the methodology. For example, there is talk of Models which are not previously named in the introduction or the method.
Response: Thank you for these suggestions, we have added some further detail into the analysis section under the methods to talk about the modeling. We have stated that we have used the forward selection method which involves using several models testing the addition of each independent variable on the dependent variable (health status) and repeating the process until there is no improvement with statistical significance. With this method non-significant predictors are excluded from the models.
Discussion: It needs to be improved. Only two articles from the last five years are discussed; the rest are not original articles or are more than five years old.
Response: As previously mentioned, this article is a brief report and to the very best of our knowledge there are no other UK studies specifically in relation to health status with cancer survivors post-treatment. We have stated this in the discussion. Therefore, this brief report is serving to develop the evidence where there is a notable gap. We have cited the two American health status studies by Weaver et al as these are the closest that we know, and we do agree that these are now somewhat old, but it is important to compare our findings with the extant literature given the gap in the evidence, regardless of the publication date. In the discussion, we do cite one of our own papers here on cancer-related self-efficacy that was published in 2022 in the Journal of Rural Health. Furthermore, we cite another Australian study by Gunn et al in the discussion that was published in 2020 which we use as a means of interpreting our findings.
Likewise, it is scarce compared to many results presented.
Response: Again, we do appreciate your comment, but the article has been submitted as a brief report and not a full research article. It is intended to provide a succinct overview of our findings in relation to health status and rural-urban residence and to stimulate further interest and work in this area. We have now added another concluding paragraph to the discussion which has increased the length of this section but again please do appreciate that this is intended to be a brief report type article.
We also received comments from one of the editors from the last round of peer review and have used these to reshape and change our article.
It is suggested to use subtitles.
Response: Thank you, we have now added subtitles into the discussion.
Conclusions: They are not clear. What is the real contribution of this research to Nursing, and if it responds to the stated objective?
Response: Thank you, a good suggestion, we have added some additional sentences to the conclusion in relation to the contribution to nursing and cancer care. We have suggested that despite our positive findings in relation to health status, it is critical that rural oncology providers do not rely on cancer survivors to co-ordinate their own care and that rural cancer nurses work collaboratively with survivors and other cancer support professionals at all stages of the pathway to facilitate a positive patient experience that will support their recovery.
Reviewer 2 Report
The paper outlining urban : rural differences in the health status of cancer survivors is of interest and could be useful in other conditions when designing interventions to improve management and coping with chronic diseases.
The methodology has some gaps and the presentation of the statistics could be more clearly outlined. In Table 1 It would be good to include the confidence intervals to see clearly the significant differences between the groups. The authors stated that the percentages may not always add to 100 because of missing data and multiple selection options , however for gender are those row or column percentages . this was not clear even with missing data and multiple options from the table.
The response rate was much lower than anticipated , however in the discussion although mentioned , there was no indication in the results that there was any adjustment in the analyses to account for non response.

Author Response
The paper outlining urban : rural differences in the health status of cancer survivors is of interest and could be useful in other conditions when designing interventions to improve management and coping with chronic diseases.
Response: Thank you for taking the time to review our article and your positive comments, we also believe that the findings could be applicable with other long-term conditions.
The methodology has some gaps and the presentation of the statistics could be more clearly outlined. In Table 1 It would be good to include the confidence intervals to see clearly the significant differences between the groups. The authors stated that the percentages may not always add to 100 because of missing data and multiple selection options, however for gender are those row or column percentages. this was not clear even with missing data and multiple options from the table.
Response: The table is intended to simply present an overview of the sample and not any form of bivariate analysis with mean differences and confidence intervals. Given the categorical nature of the variables we did use Pearson’s X to test for significance between the demographic and categorical variables and rural-urban residency. This revealed that living arrangement, marital status and qualifications were all significantly associated with rural-urban residence and we report on this in the text.
In relation to the methodology being clearer and the other reviewers comments we have now added some additional text throughout that we hope helps as well as some further detail under the analysis section in relation to the methods and models.
Apologies about the percentages not being clear. These are column percentages that we have reported, and we have added a note under the table to highlight this.
The response rate was much lower than anticipated, however in the discussion although mentioned, there was no indication in the results that there was any adjustment in the analyses to account for non-response.
Response: yes, thank you, we did not have access to the data of non-responders which could have made this challenging given we did not know the demographic makeup and characteristics of the 607 people who chose not to complete and return a survey or even all of those within the region who met the eligibility criteria. We have added some text around this in the limitations in the discussion. In sum, we could not adjust to bring the sample population in line with the total population, although as mentioned the rural-urban split was even which merited a strong comparative analysis in line with the study objective.